# TOPAS-Tissue: A Framework for the Simulation of the Biological Response to Ionizing Radiation at the Multi-Cellular Level

**DOI:** 10.3390/ijms251810061

**Published:** 2024-09-19

**Authors:** Omar Rodrigo García García, Ramon Ortiz, Eduardo Moreno-Barbosa, Naoki D-Kondo, Bruce Faddegon, Jose Ramos-Méndez

**Affiliations:** 1Facultad de Ciencias Físico Matemáticas, Benemérita Universidad Autónoma de Puebla, Puebla 72000, Mexico; omar.garciagarcia@alumno.buap.mx (O.R.G.G.); eduardo.morenoba@correo.buap.mx (E.M.-B.); 2Department of Radiation Oncology, University of California San Francisco, San Francisco, CA 94115, USA; ramon.ortizcatalan@ucsf.edu (R.O.); jorgenaoki.dominguezkondo@ucsf.edu (N.D.-K.); bruce.faddegon@ucsf.edu (B.F.)

**Keywords:** Monte Carlo track-structure simulation, agent-based modeling, multiscale modeling, radiation damage response, PC-3 cell survival

## Abstract

This work aims to develop and validate a framework for the multiscale simulation of the biological response to ionizing radiation in a population of cells forming a tissue. We present TOPAS-Tissue, a framework to allow coupling two Monte Carlo (MC) codes: TOPAS with the TOPAS-nBio extension, capable of handling the track-structure simulation and subsequent chemistry, and CompuCell3D, an agent-based model simulator for biological and environmental behavior of a population of cells. We verified the implementation by simulating the experimental conditions for a clonogenic survival assay of a 2-D PC-3 cell culture model (10 cells in 10,000 µm^2^) irradiated by MV X-rays at several absorbed dose values from 0–8 Gy. The simulation considered cell growth and division, irradiation, DSB induction, DNA repair, and cellular response. The survival was obtained by counting the number of colonies, defined as a surviving primary (or seeded) cell with progeny, at 2.7 simulated days after irradiation. DNA repair was simulated with an MC implementation of the two-lesion kinetic model and the cell response with a p53 protein-pulse model. The simulated survival curve followed the theoretical linear–quadratic response with dose. The fitted coefficients α = 0.280 ± 0.025/Gy and β = 0.042 ± 0.006/Gy^2^ agreed with published experimental data within two standard deviations. TOPAS-Tissue extends previous works by simulating in an end-to-end way the effects of radiation in a cell population, from irradiation and DNA damage leading to the cell fate. In conclusion, TOPAS-Tissue offers an extensible all-in-one simulation framework that successfully couples Compucell3D and TOPAS for multiscale simulation of the biological response to radiation.

## 1. Introduction

Monte Carlo (MC) track-structure simulations are essential in the study of radiation damage to biological matter (see, e.g., [1]). They provide a detailed description of the early physical and chemical stages of the radiation transport through matter on an event-by-event basis. Complementary to in vitro experiments, such simulations provide details of the individual processes that are challenging to measure when the required experimental spatial–temporal resolution is not achieved due to the subcellular scale. 

For decades, MC track-structure codes have been used to simulate the different aspects of the physical and chemical effects of ionizing radiation at the subcellular level. Such codes include MOCA8b [2], DBREAK [3], PARTRAC [4], KURBUK [5], Geant4-DNA [6,7], RITRACKS [8], DaMaRiS [9], and TOPAS-nBio [1]. These codes have been developed to simulate DNA damage under different irradiation conditions and to explore the spatial structure of DNA strand breaks and DNA repair. The main target of ionizing radiation in biological tissue is the nuclear DNA. It can produce a variety of lesions, including DNA Double Strand Breaks (DSBs), defined as two DNA Single Strand Breaks (SSBs) produced on opposite strands of the DNA separated by less than 10 base pairs [5]. DSBs are difficult to repair by the cell. When DSBs accumulate, unrepaired or mis-repaired, they can lead to gene mutations and chromosomal aberrations which eventually lead to cellular death [10,11].

The breaks can be produced directly when ionizing radiation deposits energy on the DNA molecule, producing physical damage [12], or indirectly, when free radicals are produced in the medium that can react with the DNA components and produce chemical damage [3,13]. The initial distribution of lesions depends on the Lineal Energy Transfer (LET) of the radiation used, which is a parameter that characterizes the density of ionizations on the medium that the radiation is traversing [2,5]. 

Previous works have reported the spatial distribution of DNA breaks induced by direct and indirect actions of radiation in geometrical models of DNA, ranging from DNA plasmids, chromatin fibers, and whole cell nucleus, see [4,6,12,14,15,16,17,18,19] (and references therein). Subsequent DNA repair in MC can be simulated with, e.g., the Two-Lesion Kinetics (TLK) model [20] in which fatally mis-repaired DSBs produce lethal lesions that accumulate as an inevitable side effect of the DNA repair [21,22]. The TLK model has been implemented in Geant4-DNA to calculate the residual number of DNA breaks and the cell surviving fraction [23]. Another repair model implemented in Geant4-DNA is proposed by Belov O., et al. (2015) [24], in which the explicit action of repair proteins is handled by a kinetic biochemical reaction scheme capable of reproducing the gamma-H2A histone family member X (*γ*-H2AX) foci fluorescence curves as DNA is repaired [25]. Cellular outcome modeling after unrepaired DNA damage accumulation, linked to the production of chromosomal aberrations [10,11] and death by mitotic catastrophe, has also been reported [21]. However, there is still a modeling gap between the DNA damage, its response, and cellular outcome. For example, no kinetic simulation was performed of biological processes such as the trigger of biochemical signals between proteins that can lead to apoptosis or other cell inactivation pathways [26,27,28,29]. In part, this can be attributed to the fact that MC track-structure simulations require a significant computational effort (weeks or months) to achieve the desired statistical uncertainty of <1% (one standard deviation) at the nanometer scale. As a consequence, the computation time is prohibitive for a cell population scenario due to its spatial dimensions.

On the other hand, agent-based modeling (ABM) codes have been used to simulate the behavior of complex cellular systems from the bottom up, assigning rules on individual cells, treated as “agents”, to interact between themselves and their environment. The response of the entire system results from the combined contributions of the individual agents. ABM has been used in multiple contexts, from epidemiology to cellular automata [30]. For research in cellular biology, ABM has been used to simulate mechanistically the behaviors of individual cells to entire populations [31]. 

The detail and scale of ABM depends on the number of agents that represent each cell. For example, in VCell software [32], an individual cell can be constructed with a large set of agents that represent several sub-cellular compartments, achieving high geometrical resolution. In contrast, codes like Biocellion [33] can simulate a vast number of cells (~10^6^) to reach the tissue scale using a single agent per cell. To accomplish such a number, the cell geometry is simplified to generic shapes such as spheres or cylinders. Compucell3D software [34,35] provides a scale in between. In this software, cells are constructed by a moderate number of agents selected from a compromise of resolution and computational speed. In Compucell3D, each cell geometry model is dynamic with diverse topologies and can consider several biological processes explicitly simulated with stochastic methods.

Neither MC track structure nor ABM codes are tailored for the complete simulation of the action of radiation in biological tissues, from the early physical interactions to long-term biological responses. Existing ABM codes are not designed to simulate the interaction of radiation with matter, and MC track-structure simulations take too much time to simulate cell populations. Therefore, the specialized functions of MC track structure and ABM approaches are complementary. A multiscale platform that couples both types of code was developed by Lui R., et al. (2021) [36]. In Lui et al., a vascular tumor model was simulated with CompuCell3D, and the geometry was translated to the Geant4-DNA application RADCELL to simulate irradiation. In their platform, the cell’s nucleus and cytoplasm were represented by two concentric spheres. This geometrical approximation was reasonably accurate for scoring absorbed dose in the cell’s compartments [36,37]. After cell irradiation, SSBs and DSBs were quantified with the Density-Based Spatial Clustering of Applications with Noise (DBSCAN) clustering algorithm [38]. Three cell states were considered: arrested, healthy, and dead. The transition between the three states was handled by random sampling of a probability distribution, where parameters were adjusted with biological data. Nevertheless, DNA repair and the explicit cell damage response were not simulated. In this way, the computational simulation was efficient, whereas the authors remarked on the extension capabilities of their platform for the implementation of external models in future works. 

The biological response to radiation involves intricate networks of proteins that control the intra- and extra-cellular processes, which are determinants of the outcome of damaged cells. For instance, the p53 tumor control protein has been identified as the central component of biological radiation response [26,28,29]. When DNA is damaged, the Ataxia Telangiectasia Mutated (ATM) protein activates, responding to DSB-repair sites from both the Non-Homologous End Joining (NHEJ) and Homologous Recombination (HR) pathways [39]. ATM initiates a chain of protein reactions that result in a series of discrete p53-pulsed signals while DNA is being repaired. p53 protein mediates between cycle arrest, promoting survival, or apoptosis signaling, like the Caspase3 (Casp3) enzyme activation, for cells that are beyond repair [29,40]; alterations in the p53 pathways are linked to carcinogenesis and other cell abnormalities [41]. Kinetic models for this protein reaction network, coupled with DNA repair models, have been published elsewhere [42,43,44]. For Monte Carlo modeling, Hu et al. [45] presented a multiscale platform to simulate the radiation and cell response in a single-cell approach that integrates the Nanodosimetry Simulation Code (NASIC) for calculating the initial distribution of DNA lesions, followed by a DNA damage repair model based on an NHEJ pathway [46] and a model for the p53 protein network [47], including the influence on cell cycle regulation. Incorporation of such models into a coupled MC track structure–ABM framework would allow the exploration of effects from a sub-cellular scale to a multi-cellar scale while quantifying observables also measured in experimental setups.

We present TOPAS-Tissue, a framework with functionalities coded in C++ and Python (version 3), to couple two MC codes specialized in particular aspects of cell irradiation and cellular behavior: Tool for Particle Simulation (TOPAS) (version OpenTOPAS.4.0), an open-source MC code capable of simulating interaction between radiation with complex biological geometrical models [48,49], using the TOPAS-nBio (version 3.0) extension to enable track structure simulation [1]; and CompuCell3D (CC3D) (version 4.5), an open-access platform written on Python for the simulation of cellular biological behaviors [50]. The framework allows the integration of CC3D cellular models in TOPAS, processing scored physical quantities, quantifying DSBs to each cell, and inputting back to CC3D for the simulation of cell growth, DNA repair, and cellular response. This is archived by combining and complementing the capabilities of TOPAS-nBio and CC3D to simulate the initial radiation damage in detail into the biological consequences and cellular outcome. We integrate into the framework a DNA repair model based on the TLK model [20,42] coupled with a cellular response model based on the biochemical response network centered around the p53 protein [44]. We verified the framework by replicating the experimental conditions of an in vitro PC-3 cell culture irradiated by MV X-ray within the range of 0–8 Gy.

## 2. Results

### 2.1. TOPAS-Tissue: Workflow

A scheme of the implemented simulation workflow of TOPAS-Tissue is presented in Figure 1. The workflow goes through three stages: pre-irradiation, irradiation, and post-irradiation. In the pre-irradiation stage, the CC3D simulation starts creating the initial geometrical distribution and shape of the cells; cells can grow and replicate during a user-defined time limit. In the second stage (irradiation), TOPAS-Tissue converts the geometrical information from CC3D into ready-to-run TOPAS-parameter control files and sets the irradiation conditions such as dose, particle type, radiation source shape, and energy spectrum. Then, the simulation of the irradiation is performed in TOPAS. The current implementation performs the irradiation in a single CC3D Monte Carlo Step (MCS). Nevertheless, developments are ongoing to allow irradiating during an MCS interval through the TOPAS time feature [51]. In the current implementation, the number of DSBs is assigned via the absorbed dose in each cell nucleus (see Section 4.2.3). In the post-irradiation stage, growth is paused for each cell with at least one DSB, and DNA and cellular repair models start simulating. The cell growth and division pause mimic the cell cycle arrest during damage repairing [29]. A more in-depth explanation of each stage is described in the Section 4.2. 

### 2.2. Cell Culture Evolution during the Pre-Irradiation Stage 

In the pre-irradiation stage, the experimental in vitro PC-3 cell culture setup from Wakisaka Y. et al. (2023) [52] was reproduced. The initial number of randomly placed cells was 2.48 cells per 100 µm^2^ as reported for the initial experimental seeded density in [52], 4 days before irradiation. The cell colonies replicated until the confluence reached 10.2 cells per 100 µm^2^, at the irradiation time of 4.1 days after seeding, equivalent to 2500 Monte Carlo Steps (MCS), which agreed within 1.6% from the value reported in the experimental setup [52].

In addition, we determined a set of adhesion values (Table 1) that reproduced the experimental conditions providing a stable geometry model of rounded cells in sparse groups that allowed safe mobility between neighbor cells. The time evolution of the geometrical model for the cell culture is shown in Figure 2.

### 2.3. Irradiation Stage and Execution Time

The dose distribution computed with TOPAS in the cell compartments irradiated by electrons from 4 MV X-rays is shown in Figure 3A. The average number of DSBs followed a linear response with dose, with a slope of 25.1 DSBs/Gy, as shown in Figure 3B. The computation time per CPU for the execution of the TOPAS simulation, T_TOPAS_, scaled linearly with the number of simulated electrons, and consequently, with the dose D, following T_TOPAS_ = 2.9 h·D/Gy + 1.4 h. On the other hand, CC3D execution time remained almost constant between 1 Gy to 4 Gy (9.13 h) and decreased linearly from 4 Gy to 8 Gy with a slope of −0.55 h/Gy, as shown in Figure 3C. The reduction of computing time was caused by the decrease in the number of simulated cells due to the increase in killing with the increase in dose. 

### 2.4. Cell-Survival Curve

Results from the multiscale simulation of cell survival as a function of the dose are shown in Figure 4. Each point corresponds to individual simulations at specific radiation doses. Error bars represent statistical uncertainties, measuring two standard deviations. Experimental values from [52] for a 4 MV X-ray beam are also displayed. As depicted, the simulated curve followed an exponential response. We fitted a linear–quadratic exponential function, obtaining parameters α = 0.280 ± 0.025/Gy and β = 0.042 ± 0.006/Gy^2^. These values agreed within one standard deviation from the experimental parameters α_0_ = 0.302 ± 0.008/Gy and β_0_ = 0.0417 ± 0.0049/Gy^2^ [52]. 

## 3. Discussion

The TOPAS-Tissue framework for the coupling of two MC-based software, TOPAS and CompuCell3D, was implemented and verified. The framework allowed the multiscale simulation of cell survival from irradiation to the biological impact on cellular outcome. Verification included a comparison between simulated and measured cell survival of PC-3 cells under X-ray irradiation.

CC3D, managed by TOPAS-Tissue, was able to reproduce the evolution of the cell culture during the pre-irradiation stage, archiving a similar value for the final cell density at the same time period reported in the experiment. Since the setup is based on an in vitro assay, we assumed optimal conditions for the cell culture in this first approach. In particular, all cell stages were synchronized, and cell growth followed a constant rate. However, these and further factors linked to the cell cycle alter biochemical conditions, such as biochemical signaling in the cellular environment and oxygen availability [53], affecting the cell’s development and radiosensitivity [54]. To incorporate these considerations in TOPAS-Tissue, ongoing integration of the PhenoCellPy [55] package for CC3D will follow for improving the modeling detail of biological behavior. In addition, the effect of oxygen or the biochemical signaling to neighborhood cells can be modeled with CC3D chemical field diffusion tools. These tools can be used to simulate the Prostaglandin E2 (PGE2) pro-mitotic factor that is produced by Casp3 on dying cancer cells and stimulates the repopulation of surviving cells [56]. CC3D functions also allow for a growth rate that is influenced by the concentration of such signals [50,53]. In this regard, the perturbation of oxygen fields by physical–chemical parameters, e.g., dose rate, can be precomputed and incorporated into simulations through TOPAS-Tissue look-up tables; for applications in FLASH radiotherapy, see [17].

There are several factors that influence the cell radiosensitivity, such as oxygen availability, nutrient gradients, and mechanical stress. CC3D has tools to handle all these factors as diffusive chemical fields. In this way, oxygen and nutrient distribution and cellular intake–uptake can be accounted for [53]. Mechanical stress is simulated with the Potts algorithm of CC3D by, e.g., automatically tracking quantities like the internal pressure of each cell, which is directly related to the mechanical stress [50].

In this work, the DSB assignment to individual cells was derived from the absorbed dose to each cell nucleus. This approach is a reasonable estimate for MV X-ray irradiation considering the homogeneity of energy deposition events by this radiation quality. However, the lack of spatial detail distribution of complex damages through the cells must be considered for other particle types for which the energy is more densely distributed along their primary track, e.g., proton or carbon ions. Ongoing implementations in TOPAS-Tissue include alternative methods for DNA damage assignment. For example, clustering algorithms including DBSCAN are available in TOPAS-nBio for the damage distribution calculation [12,38] and were used in RADCELL via Geant4-DNA, which was the first work in reporting the interface between CC3D and Geant4-DNA. In addition, precomputed lookup tables that relate accumulated micro- and nano-dosimetric quantities with the probability of lesion inductions and their spatial distribution have been proposed [57]. TOPAS-Tissue has the flexibility and extensibility to compute other relevant physical and chemical quantities through different TOPAS and TOPAS-nBio scorers to directly calculate the number of DSBs and their complexity from the incident radiation.

The TLK model used in this work considers two repair pathways corresponding to slow and fast kinetics. This is a simplification of the actual repair pathways, as it does not consider explicitly the reactions of the involved proteins. Therefore, cells with deficient protein expressions, like the XPF-deficient human fibroblasts (XFE), cannot be simulated with this model. Models that consider an extended set of protein reactions have been reported elsewhere [24]. However, the pathway for cell fate in the previous work is not considered.

In this work, the cell fate is simulated with a p53 protein network model that captures the general dynamics of the biological response to radiation. Mutations on the p53 pathway, like the ones on the *TP53* gene sensory layer that influence the DNA repair process activation [58], and other cell behaviors induced by radiation like cell cycle regulation, such as the phosphorylation of the checkpoint kinases (Chk1 and Chk2) by ATM [56,59], are not considered. These key players are important for the simulation of normal cells. A model describing the effect on cell cycle regulation, including the action of Chk1 on cell arrest, has been reported elsewhere [60]. This and other models to simulate distinct cell behaviors can be added to TOPAS-tissue in SBML format through CC3D.

The simulated results for the survival curve agreed with experimental data within two standard deviations, validating the performance of TOPAS-Tissue and the models implemented. However, the current implementation of the cell response only considered one cell death mechanism (apoptosis) integrated into the p53 pulse model. Future works using TOPAS-Tissue might consider other cell fate models with different pathways, including products of mis-repaired damage such as chromosome aberrations [10,11]. In addition, a model that considers the cell cycle is needed to improve the description of the death pathways. Future work also consists of integrating the model proposed in Iwamoto et al. (2011) [60]. In that kinetic model, the protein reactions that control the cell cycle and its response to radiation damage are considered.

## 4. Materials and Methods

### 4.1. Software

#### 4.1.1. CompuCell3D

Compucell3D (version 4.5, accessed on December 2023) is a well-established code for agent-based modeling of cell behavior. It is based on the Glazier–Graner–Howeg (GGH) algorithm [61], an evolution of the Cellular Potts Model [62]. Briefly, in the Cellular Potts Model, the simulation space is discretized in identical geometrical elements or voxels. A cell is represented by a subset of these voxels identified by a unique index. The time evolution of the system is managed on a step-by-step basis using Monte Carlo Steps (MCSs). At each step, a set of voxels from every cell attempt to copy themselves to neighborhood voxels in a process called an index copy attempt. The probability of success is randomly sampled from a Boltzmann distribution in terms of the effective energy of each cell. The effective energy encompasses several constrictions on the cell’s volume, surface, adhesion to neighbor cells, or response to external chemical stimuli. Growth and mitosis can be explicitly simulated in this way, for example, by modifying the volume constriction as a function of time and establishing division conditions. CC3D can handle both the chemical diffusion of substances in the cell environment and inner biochemical networks in the individual cell’s microenvironment [63].

In the context of cancer research, this software has been employed to simulate the invasion of malignant cells into normal tissue [64], vascularized tumors, and the angiogenesis process induced by hypoxic cells on its core [53], as well as multiscale models for the evolution of a tumor’s growth under chemotherapy and the interaction between the drug and the malignant cell’s biochemical signaling network [65]

#### 4.1.2. TOPAS

The TOPAS version used in this work is OpenTOPAS v4.0 (based on Geant4 v11.1.3) available on the TOPAS collaboration GitHub (https://github.com/OpenTOPAS, accessed 10 October 2023). This version of the TOPAS code is a continuous development from TOPAS version 3.9.

TOPAS [48,49] wraps and extends the general-purpose MC toolkit Geant4 [66] via a user-friendly interface. It has been used extensively to study radiation transport in the field of medical physics. In radiation biology, its MC track-structure extension TOPAS-nBio has been validated for the simulation of the physical, pre-chemical, and chemical stages of ionizing radiation [1,67,68]. TOPAS-nBio has been used to simulate the induction of DNA lesions by radiation in plasmids under different conditions, including temperature, scavenging capacity of solutes, and DNA plasmid supercoiling grade [17,19]. For cellular geometries, TOPAS-nBio has shown full flexibility to simulate SSB and DSB yields, repair, and dicentric and acentric fragments [69], dose and ROS enhancement by gold nanoparticles [55], and nucleus damage by novel radioisotopes [70], among many other applications.

### 4.2. TOPAS-Tissue: Simulation Stages and Models

TOPAS-Tissue simulations are separated into three stages: pre-irradiation stage, irradiation stage, and post-irradiation stage, explained in more detail in the next sections. The necessary parameters input into CC3D include the space grid size, initial number of cells, equivalence between metric units and CC3D internal units (voxels to length units and MCSs to time units), cell’s volume growth rate, adhesion energy between neighbor cells and the cell’s environment, and several others shown in Table 2.

#### 4.2.1. Pre-Irradiation Cell Culture Growth Stage

In the pre-irradiation stage, CC3D was used to generate the cell population’s geometry and handle the individual cells’ biological behavior. The TOPAS-Tissue code was designed to be imported as a Python object in the initialization function of the so-called Steppable class of CC3D.

CC3D did not provide a system of units. Thus, a function to set the equivalence of unity voxels to units of length was provided in TOPAS-Tissue. This facilitated the translation of CC3D geometrical parameters to TOPAS components. The equivalence from the MCSs to units of time allowed for the synchronization of all the time-dependent processes simulated, with default parameters set as 1 voxel = 1 µm and 1 MCS = 1 min.

CC3D handled the pre-irradiation stage in which the cell population evolves from an initial number of seeded cells to a desired confluence (percentage of the surface area covered by cells). The information for the cell types, their volumes V_i_ and doubling times t_v_, were input to define the growth and division process. These values were stored in a dictionary, which was accessed and modified by TOPAS-Tissue. One central parameter for any CC3D simulation is the adhesion energy between different cells and their environment. This parameter defined the shape, mobility, and proximity to neighbors of a given cell type. In this work, we determined the optimal adhesion energy value as the value that allowed us to reconstruct a stable geometric model for the use-case described below. The geometric model was stored in Potts Initial Format (PIF), a standard data input file in CC3D simulations [63], which served as a starting point for each simulated irradiation.

#### 4.2.2. Irradiation Stage

At the irradiation stage, TOPAS-Tissue automatically transferred from CC3D to TOPAS the geometrical information in the form of a voxelated phantom using the TsImageCube component of TOPAS. The phantom dimensions were inherited from CC3D, whereas the orientation was set according to the TOPAS coordinate system. The phantom contained the cell and cell compartment identification numbers as information in each voxel, which facilitated cell identification in the postprocessing. We assumed the cell’s volume and its environment were made of water, thus the TsImageCube component allowed to set water material to all voxels independently of the voxel value. TOPAS parameter files were created automatically for the radiation source specifications (particle type, spatial distribution, angular momentum, and energy spectrum), scoring of physical quantities (e.g., absorbed dose), visualization, and simulation control. TOPAS was automatically called for a run. After the irradiation simulation, the output file was processed by TOPAS-Tissue before sending it back to CompuCell3D to continue the simulation. The output file with the dose distribution was processed to retrieve the dose on each cell’s compartment (nucleus and cytoplasm). To this end, a subroutine was provided in TOPAS-Tissue, which takes advantage of the CC3D capability to track the indices of the subset of voxels composing every cell and associate them with the indices of a TOPAS volumetric scorer.

Due to the CC3D intrinsic nature of implementing geometrical models in a grid geometry and the capabilities of TOPAS/TOPAS-nBio to generate voxelized geometries, TOPAS-Tissue offers a flexible platform to irradiate any CC3D geometrical model. Other types of cells, cell population arrangements (tissue), and their behavior can be incorporated by defining new phenotypes in CC3D. The parameters to define such phenotypes might include, the growth rate, volume, and topological shape, among many others; ongoing developments to incorporate PhenoCellPy (version 1.0) [71] shall be presented in future works.

#### 4.2.3. DSB Assignment

In a cell population irradiated with a uniform particle field at low LET (~0.2 keV/µm) radiation, a reasonable approximation to compute the number of DSBs is from a Poisson distribution, where the mean number of lesions increases linearly with the absorbed dose [72]. This approach was adopted in the current work to facilitate the coupling and speed-up the testing of the p53 network model. Nevertheless, TOPAS-Tissue is not limited to the assignment of initial DNA damage using the Poisson distribution approach. More advanced methods to compute DSBs are available in TOPAS-nBio and can be accessed through the proposed framework. For example, the DBSCAN algorithm and a whole DNA cellular nucleus model are available in the TOPAS-nBio suite of scorers, among several others [1]. Therefore, for each cell, once the dose to the nucleus D_nuc_ was acquired, DSBs were randomly sampled from a Poisson distribution with an expectation value equal to ${\bar{N}}_{DSB}$ D_nuc_. The mean number of DSBs per unit dose, ${\bar{N}}_{DSB}$, depended on the radiation quality. A ${\bar{N}}_{DSB}$ value of 27.5 DSBs/Gy was used in this work [24]. Complex DSBs (DSB_2_s) were considered given by a fraction *f* from the total number of DSBs. Consequently, simple DSBs were calculated as DSB_1_ = (1 − *f*) DSB. We used a fraction value *f* = 0.51 for X-rays, obtained from [24,72]. DSBs were assigned to each cell using a Python dictionary provided by CC3D, which allowed the use of user-defined parameters during the simulation. We used such information to dynamically input the DNA repair model during each MCS. 

#### 4.2.4. DNA Repair and Cell Response Models

Of note, the homologous rejoin and non-homologous end-join repair mechanisms perform the DNA repair and thus are not directly linked to the p53 pulses [58]. However, the model used in this study was based on the one proposed by Zhang, X.P. et al. (2011) [44], where the two lesion kinetic DNA repair model works in parallel and influences the p53 protein network through the activation of the ATM protein by the DSB_C_ complex. The DNA repair model consisted of an MC version of the TLK model for DSB rejoining [42]. In this model, individual DSBs could transition between three states: an intact DSB (DSB), a DSB attached to a repair protein complex (DSB_C_), and a DSB repaired or fixed (DSB_F_) [42,43,44]. The interest of this model was put exclusively on the DSBs being repaired (state C), since they influence the activation of certain proteins on the cellular response model, but no distinction was made on the lethality of fixed lesions on the last state [42,43,44], shown in Equation (1); although, the authors recognized that the accumulation of lethal lesions had profound consequences on the final cell viability [42]. In contrast, the original TLK model [20] separated the non-lethal fraction of lesions from the lethal ones that accumulated in the repair process. In our implementation, the DNA repair model used in Zhang’s work was modified to differentiate non-lethal DSBs (DSB_N_) from lethal lesions (DSB_L_), considered in this work as non-repairable DSBs, shown in Equation (2). The DSB_N_ occurred at a probability given by 1 − p_lethal_, where p_lethal_ (1.5% in this work) is the probability of producing a DSB_L_ (Figure 5A). This value of p_lethal_ comes from the fraction of *γ*-H2AX foci at 24 h post-irradiation, considered as the fraction of non-repairable DSBs. The experimentally measured value for 1 Gy γ-ray irradiation (LET ~ 0.2 keV/µm) was 1.35 ± 1.25% [73].
(1)$$DSB_{C}\overset{k_{fix}}{\to }DSB_{F}$$
(2)$$DSB_{C}\overset{k_{fix}}{\to }\left\{\begin{array}{l} {\mathrm{DSB}}_{N}\;(1-p_{lethal})\\ \;\\ {\mathrm{DSB}}_{L}\;p_{lethal} \end{array}\right.$$

The probabilities of transitioning between states were regulated by association–dissociation rates k_i_, the number of available repair proteins NR, and the number of DSBs in each state. In addition, the repair of simple lesions (DSB_1_) was handled by a fast kinetics component, while the repair of complex lesions (DSB_2_) was handled by a slow kinetics component of the model.

During the simulation, the DNA repair module communicated with the cell response model as follows: In the damage sensor module (Figure 5B), the number of DSBs in state C influenced the activation rate of the ATM protein (d[ATM*]/dt). This activation followed Michaelis–Menten kinetics, as follows: (3)$$\frac{d[ATM^{*}]}{dt}=kact\cdot \left[ATM^{*}\right]\frac{DSB_{C}\;}{DSB_{C}+j_{DSB_{C}}}\cdot \frac{\left[ATM\right]}{\left[ATM\right]+jact}$$
where k_act_, j_DSBc_, and j_act_ were the maximum activation rate, and thresholds for the number of repairing DSBs and the ATM concentration, respectively. Next, in the feedback control module, multiple proteins were activated and deactivated at specific rates, producing pulses in the concentration of the p53 protein (Figure 5C). Lastly, in the cell outcome decision module, if the DNA repairs on time (~30 h after five p53 pulses) the cell cycle arrest was paused; otherwise, the sustained oscillations of p53 lead to the triggering of Casp3 (Figure 5D). The coupled model was validated elsewhere for MCF7 human breast cancer cell assays treated with the radiomimetic drug doxorubicin. The parameters, reported in [44], are listed in the Appendix A and are followed by a scheme of the interaction between all the systems, presented on Appendix A. To verify the implementation of the model, the time evolution curves of the concentration, its amplitude and period from p53, ATM*, and Wip1 pulses were compared with those from [44], a comparison between the TLK and MC repair models was performed as well, results are shown in Appendix A. Experimentally it was reported that a maximum of 5 pulses was observed for apoptotic cells over a period of 6 h between oscillations [27,29].

We implemented the coupled model in the Systems Biology Markup Language (SBML) format [74] using Moccasin software (version 1.3.0) [75]. CC3D has the capability to read models in SBML format and assign them to each individual cell [63,76]. To synchronize the cell population growth, the DNA repair, and cell response models, an equivalence of 1 MSC equal to 0.5 min was determined. The simulation was run by 8000 MCSs, equivalent to 2.7 days after irradiation. The time limit was enough to allow every cell to finish its DNA repair or death process [44] and perform at least one cell division.

#### 4.2.5. Cell Survival Quantification

The two individual conditions to evaluate cell death were considered per cell: the presence of at least one lethal lesion and the triggering of a Casp3 signal. In the absence of DNA damage or complete DNA repairing, the Casp3 level remained close to its initial value of 0.05 µM. However, once it was triggered by p53 sustained oscillations, it rapidly increased and saturated at a concentration of 2.7 µM (Figure 5D). We set this value as the threshold to consider the cells dead for apoptosis. The Casp3 level was checked at every MCS while the cell’s DNA DSB(s) was being repaired. The presence of lethal lesions was checked at the end of the DNA repair. As part of the apoptotic process, the cells reduced their volume [77]. To simulate this process, the loss of volume until cell disappearance was modeled as a constant rate [53] of −1.54 µm^3^/min, which corresponded to the negative of the growth rate. On the other hand, if all the DSBs of a cell were repaired, the cell survived and returned to its proliferative state.

The simulation tracked internally the information for every cell at each MCS, including the number of DSBs, the concentration of p53 and Casp3, and a colony ID number. We assigned a unique colony ID to each primary seeded cell at the time of irradiation as an attribute. After cell division, the clones (daughter cells) were assigned the same colony ID as the primary (mother) cell. This attribute allowed colony identification and counting. The time evolution of the number of cells was scored at user-defined steps (100 MCSs by default). The end of the simulation was dictated by the MCS at which the kinetic models have reached equilibrium (~30 h after irradiation). After this time, the survival was quantified from the number of remaining colonies as a function of the absorbed dose at 66 h, corresponding to 2 division cycles after irradiation.

### 4.3. Simulation Setup

The input parameter values used in the simulation framework are shown in Table 3. A detailed description of the culture and radiation source models is presented in the following sections.

#### 4.3.1. PC-3 Cell Culture Phantom Geometric Model

A PC-3 cell culture phantom was constructed in CC3D using a 1000 × 1000 µm^2^ surface lattice dimension and 60 µm depth. The lattice was thick enough to avoid collisions between the cells and the phantom upper edge. The average PC-3 cell size was reported as 18.08 ± 2.69 µm in diameter [78]; therefore, a cell radius of $r_{PC-3}$ = 9 µm was selected, yielding a volume of 3053 µm^3^. The average cellular nucleus radius was reported to be between 3 to 5 µm [79]. Therefore, in this work, a nuclear radius r_nuc_ = 4 µm was used for all the cells. The lattice equivalence between a cubic voxel side size and length in µm was 1:1. The geometry also included a layer of one voxel thickness placed at the bottom of the phantom to represent the interface between the flask and the cells (see Figure 6C). In the pre-irradiation stage, the equivalence between the MCS and the time step was set to 1 day = 500 MCSs, and the cell’s growth rate was 3053 voxels every 687 MCSs. In this way, the cell doubling time agreed with the reported value of 33 h [52]. At the beginning of the simulation, to achieve the desired confluence, two to three cells were randomly placed in subregions of 100 × 100 µm^2^ within the phantom. The simulation was run to simulate 4.1 days (2050 MCSs) of cell division. Cells were divided in random directions in the X–Y direction (Figure 6C). For the post-irradiation stage, the time equivalence was changed to 1 MCS = 0.5 min, and as a consequence, the growth rate changed to 3053 voxels every 3960 MCSs (0.77 vox/MCS).

The default parameters for the cell adhesion, combined with the fact that CC3D does not have the option to control the adhesion to the lattice frontier, produced cells accumulated in squared bulks. Therefore, optimal adhesion parameters were determined by a series of multiple simulations (see Table 1 in Results Section 2.2).

#### 4.3.2. X-ray Irradiation

The phase space file from a 6 MV TrueBeam LINAC was obtained from MyVarian at www.myvarian.com (accessed 30 November 2023). The phase space (10^9^ histories) was used in TOPAS to irradiate a water phantom of 10 cm × 10 cm × 20 cm placed with a 5 × 5 cm^2^ field at 100 cm surface-to-surface distance. A scoring phase space plane was placed at a depth of 10 cm; a region with transient charged-particle equilibrium. The kinetic energy at the vertex position from only the secondary electrons reaching the plane was scored. The TOPAS physics module was “g4em-standard_opt4” with a production cut for secondary electrons of 1 µm. Later, the kinetic energy spectrum was used to construct a volumetric and isotropic electron source uniformly distributed within the voxelated phantom containing the cell culture geometry. On average, 18.2 million source electrons were needed to retrieve a mean absorbed dose to all the cell nuclei of 0.994 ± 0.001 Gy. TOPAS-tissue is compatible with the TOPAS-nBio Monte Carlo track structure. In this way, the SSBs and DSBs required by the DNA model repair (Section 4.2.3) can be simulated with TOPAS-nBio for a large set of particles and energies, including microdosimetric and nanodosimetric quantities [80].

Results from 20 simulations using different random seeds run using a Dell Precision 5820 Tower 20-CPU Intel Xeon W-2155 3.30 Ghz processor (Round Rock, TX, USA) were performed.

## 5. Conclusions

A new framework for the multiscale simulation of the irradiation and biological evolution of a cell population was developed. The framework called TOPAS-Tissue couples two well-established Monte Carlo-based codes: TOPAS and CC3D. One specializes in the simulation of radiation transport and its interaction with matter and the other in the simulation of the cell’s biological behaviors. The framework was used to construct a model of a PC-3 cell culture irradiated with an MV X-ray source in the range of 0–8 Gy. Our framework allowed an easy simulation setup to reproduce the pre-irradiation growth of the cell culture from an initial number of seeded cells to a final confluency. The dose was scored in two distinct cellular compartments: cytoplasm and nucleus. DSBs were randomly sampled, following a Poisson distribution, from the dose in each cell’s nucleus, and assigned. A coupled model consisting of DNA repair and a cell response model were implemented. Two cell inactivation conditions were considered: the presence of lethal lesions and apoptosis triggered by the Caspase3 enzyme signal. The cell survival curve was simulated from the counting of colonies. The parameters from a fitted linear quadratic model differed by two standard deviations from published measured data.

The new framework offers an all-in-one multiscale platform for the irradiation of multi-cellular structures and their biological response modeling. It considers the spatial distribution of radiation-induced lesions, changes in the cellular environment, such as oxygen availability, and models for the radiation influence on internal cellular processes like cell cycle regulation, which is an ongoing work for future publications. Overall, this work provides the foundation to assist future research on the understanding of tissue response to radiation, which is crucial for the development of new radiotherapy techniques, by enabling coupled and versatile simulations of physical and biological processes using two dedicated and validated softwares.

## Figures and Tables

**Figure 1 ijms-25-10061-f001:**
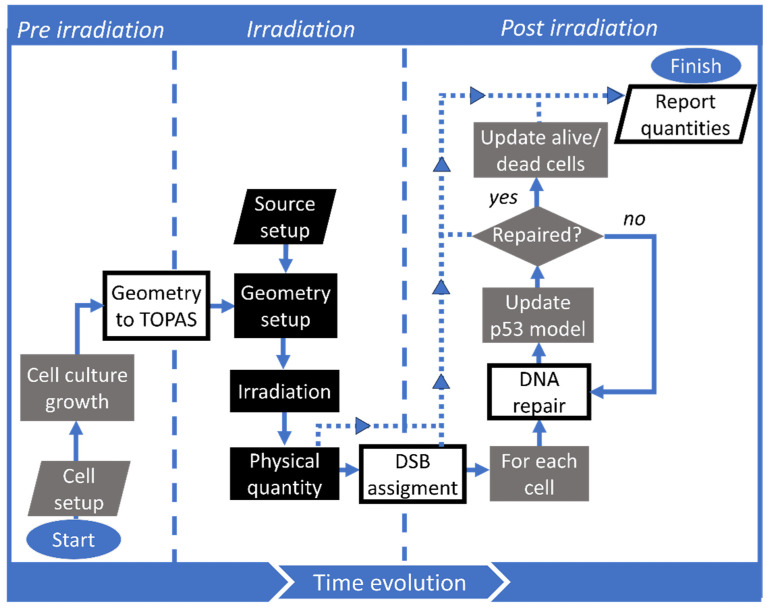
TOPAS-Tissue workflow showing the three stages of the simulation handled by CC3D (gray), TOPAS (black), and TOPAS-Tissue (white). Stages are delimited with vertical dashed lines. Arrows with solid lines indicate the process flow, while with dotted lines indicate the reported quantities.

**Figure 2 ijms-25-10061-f002:**
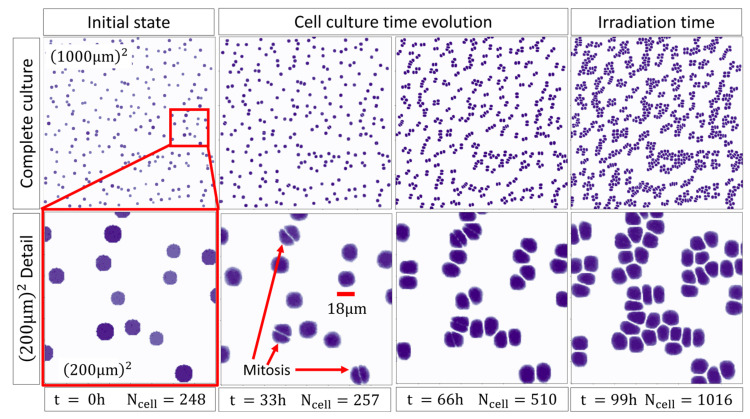
2D projection of the cell culture time evolution in the pre-irradiation stage. Red squares represent a zoom on a (200 µm)^2^ region of the whole culture, red arrows point to the first cells to undergo mitosis. The red scaling bar is 18 µm length, which is the average diameter of the PC-3 cells.

**Figure 3 ijms-25-10061-f003:**
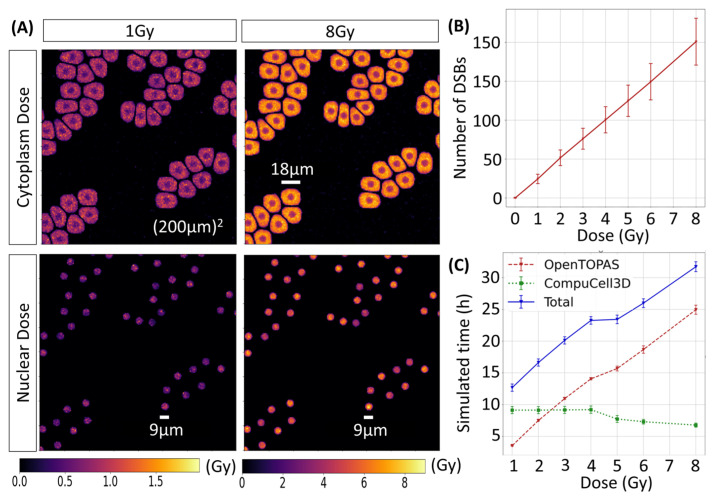
(**A**) 2D projection of the dose distribution on the cytoplasm and nucleus at 1 and 8 Gy. (**B**) The number of DSBs as a function of the dose. (**C**) Simulation time for TOPAS and CC3D as a function of the dose.

**Figure 4 ijms-25-10061-f004:**
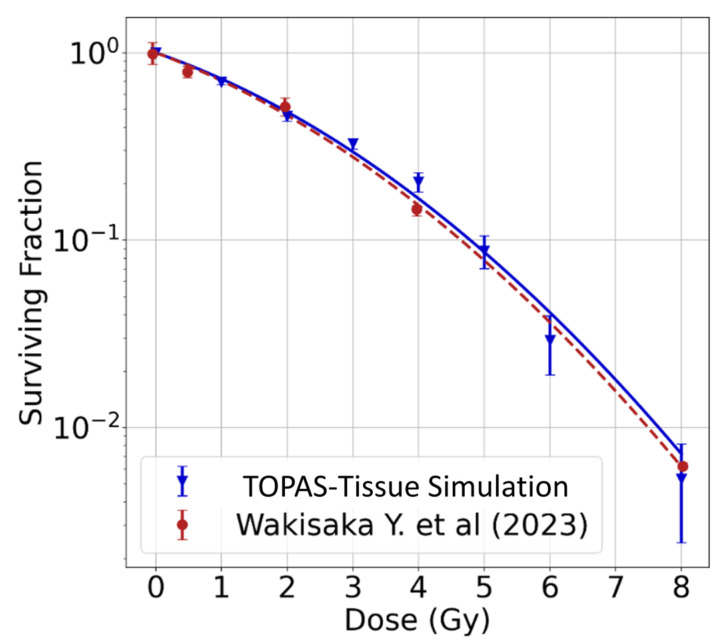
Simulated survival fraction (triangles) compared with experimental data (circles) for MV X-rays from Wakisaka et al. (2023) [52]. Lines are the fitting of the linear–quadratic model.

**Figure 5 ijms-25-10061-f005:**
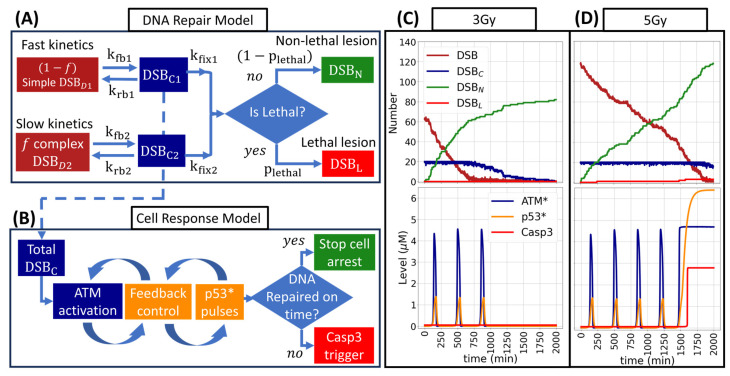
(**A**) DNA repair model. (**B**) Scheme for the cell response model. (**C**,**D**) Time evolution of DNA repair kinetics at 3 Gy (**C**) and 5 Gy (**D**); time evolution of p53 pulses is shown in the bottom panels. For the 3 Gy, DNA was repaired on time and no lethal DSBs were produced, leading to cell survival. For 5 Gy, lethal lesions were produced, and the sustained oscillation of p53 triggered Casp3, leading to cell death. Arrows with solid lines represent the process flow, while the dashed line represents the influence by the repair model on the cell response model through the total number of DSB_C_. The color scheme on the graphs (**C**,**D**) match the corresponding box states on panels (**A**,**B**).

**Figure 6 ijms-25-10061-f006:**
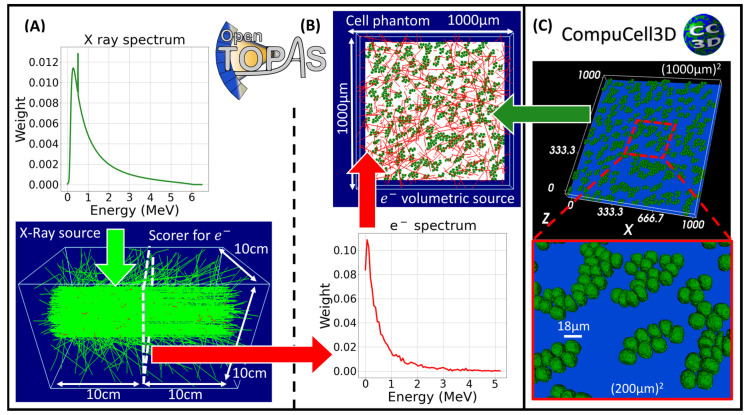
Irradiation scheme using TOPAS: (**A**) A Varian 6 MV X-Ray phase space (spectrum in the top panel of (**A**) and green lines are photon tracks on the bottom panel) is used to retrieve the kinetic energy spectrum from the electrons produced at a depth of 10 cm as represented by the red arrow. (**B**) Then a volumetric electron source (spectrum in the bottom panel of (**B**) and red lines are electron tracks on the top panel) is assigned to the phantom geometric component containing (**C**) the geometrical information from the CC3D’s cell culture. Transfer of information is represented by the green arrow; the red squares represent a zoom on a (200 µm)^2^ region of the complete culture.

**Table 1 ijms-25-10061-t001:** Adhesion parameters between PC-3 cells, their environment, and the flask interface.

Type	Medium	Flask	Alive PC-3	Dead PC-3
Medium	0.0	0.0	1.0	10.0
Flask	-	0.0	100.0	100.0
Alive PC-3	-	-	10.0	1.0
Dead PC-3	-	-	-	100.0

**Table 2 ijms-25-10061-t002:** List of the user tunable parameters and a brief description for each one.

Parameter	Description	Parameter	Description
$X_{max}$	Number of voxels on the X direction	$t_{doub}$	Cell’s doubling volume time
$Y_{max}$	Number of voxels on the Y direction	$k_{growth}$	Cell’s growth rate
$Z_{max}$	Number of voxels on the Z direction	$T_{pre}$	Pre-irradiation period duration
$vox_{eq}$	Equivalence from voxels to length units	${\bar{N}}_{DSB}$	Average number of DSBs per Gy
dt	Equivalence from MCSs to time units	$N_{p}$	Number of primary particles
$r_{cell}$	Cell radius	D	Prescribed dose to the phantom
$V_{cell}$	Cell volume	f	Fraction of complex DSBs
$r_{nuc}$	Nuclear radius		

**Table 3 ijms-25-10061-t003:** Values for the parameters used in the simulation’s configuration.

Parameter	CC3D Units	Metric Value	Parameter	CC3D Units	Metric Value
$X_{max}$	1000 vox	1000 µm	1−f	-------------------	49%
$Y_{max}$	1000 vox	1000 µm	$p_{fatal}$	-------------------	1.5%
$Z_{max}$	60 vox	60 µm			
$vox_{eq}$	1 vox	1 µm	Pre-irradiation stage		
$r_{PC-3}$	9 vox	9 µm	dt	1 MCS	2.88 min
$V_{PC-3}$	3053 vox	3053 µm^3^	$T_{pre}$	2050 MCSs	4.1 days
$r_{nuc}$	4 vox	4 µm	$t_{doub}$	687 MCSs	33 h
${\bar{N}}_{DSB}$	-----------------	27.5 DSB/Gy	$k_{growth}$	4.44 vox/MCS	$1.54\;\mu m^{3}/min$
D	-----------------	0–8 Gy	Post-irradiation stage		
$N_{e}^{-}$	-----------------	$18.2\times 10^{6}$/Gy	dt	1 MCS	0.5 min
f	-----------------	51%	$t_{doub}$	3960 MCSs	33 h
			$k_{growth}$	0.77 vox/MCS	$1.54\;\mu m^{3}/min$

## Data Availability

The original contributions presented in the study are included in the article/Appendix A; further inquiries can be directed to the corresponding author.

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
