# Peer review of "TOPAS-Tissue: A Framework for the Simulation of the Biological Response to Ionizing Radiation at the Multi-Cellular Level"

_ijms, 2024, doi:10.3390/ijms251810061_

Round 1
Reviewer 1 Report
Comments and Suggestions for Authors
This is an interesting article that proposes a new method coupling TOPAS-nBio for track structure simulation and CompuCell3D for simulating cell population behavior. It provides an important tool for studying the problem of how radiobiology DNA scale effects are combined with macroscale effects. However, this reviewer believes that substantial revisions to enhance the significance of this work are necessary before consideration for publication.
Major points
1. In INTRODUCTION, the most relevant studies are not discussed, for example,
[1] Dolan D W, Zupanic A, Nelson G, Hall P, Miwa S, Kirkwood T B and Shanley D P, 2015. Integrated Stochastic Model of DNA Damage Repair by Non-homologous End Joining and p53/p21-Mediated Early Senescence Signalling. PLoS Comput Biol, 11, e1004246.
[2] Hu, Ankang et al.“Modeling of DNA Damage Repair and Cell Response in Relation to p53 System Exposed to Ionizing Radiation.” International journal of molecular sciences vol.23,19 11323. 26 Sep.2022, doi:10.3390/ijms231911323.
[3] Hat B, Kochanczyk M, Bogdal M N and Lipniacki T, 2016. Feedbacks, Bifurcations, and Cell Fate Decision-Making in the p53 System. PLoS Comput Biol, 12, e1004787.
2. In 2.2.3. DSB assignment, the estimation of DSB is slightly rough and only applies to low LET, the expectation of DSB number depends only on the nuclear dose, the literature-based [59] is old, is it possible to obtain more advanced methods (e.g., DNA-model-based track structure simulation) and apply on more cases?
3. To support the significance of the study, RESULT needs more information and verification about p53 and the DNA repair module, namely, discussions on how p53 and DNA repair affect the result of simulation are suggested be supplemented.
4. The cell culture growth in this paper seems to be mainly aimed at in-vitro culture, there is still a certain gap with human tissues. We expect the TOPAS-Tissue framework to be further mature, for example, considering in-vivo tissue environment and cell interactions, so as to truly expand to the tissue level effect in the future.
Minor points
1. Workflow is not suggested to be placed in RESULT.
Author Response
Thank you for taking the time to review this manuscript. Please see the attachment for the full respones

Reviewer 2 Report
Comments and Suggestions for Authors
This paper presents TOPAS-Tissue, a framework that couples Monte Carlo codes TOPAS (with TOPAS-nBio) and CompuCell3D to simulate the multiscale biological response of cell populations to ionizing radiation. The framework accurately models cell growth, DNA damage and repair, and cellular responses, verified through simulations of a 2-D PC-3 cell culture irradiated by MV X-rays. The survival curve closely matched experimental data. However, the paper requires major revisions for clarity, a detailed explanation of the methodology, and more comprehensive validation against diverse experimental conditions.
Comment for authors
1. Indicate the novelty of this work compared to existing literature in abstract
2. The introduction section lacks sufficient background information and requires significant improvements. It should include a recent literature review on the biological effects of radiation for new readers.
3. Add information regarding the interaction of radiations with the biological systems for new readers. Incorporation of the following recent article in the introduction section might be helpful [https://doi.org/10.3390/ijms23169288].
4. How does TOPAS-Tissue perform with other cell types or 3-D cell cultures beyond the PC-3 model?
6. Provide a more detailed explanation of the parameter selection process for the DNA repair and cellular response models, particularly the p53 pulse model and the two-lesion kinetic model?
7. What are the limitations and assumptions inherent in the current DNA repair and cell response models, and how do these assumptions impact the simulation results?
8. How does the framework handle different types of ionizing radiation, such as protons or heavy ions, which have different energy deposition patterns? Discuss in the manuscript.
9. How does the framework integrate external factors influencing cell behavior, such as oxygen availability, nutrient gradients, and mechanical stress, which can affect radiosensitivity?
10. The paper contained typos and grammatical errors. Double-check and correct them in the revised version.
Comments on the Quality of English LanguageThe paper contained typos and grammatical errors. Double-check and correct them in the revised version.
Author Response

(The authors gave the same response as above.)

Round 2
Reviewer 2 Report
Comments and Suggestions for Authors
The authors have addressed my comments in the revised version. I recommend accepting the paper for publication.